# Hepatobiliary phase signal intensity: A potential method of diagnosing HCC with atypical imaging features among LR-M observations

Jae Hyon Park, Yong Eun Chung◉*, Nieun Seo, Jin-Young Choi, Mi-Suk Park, Myeong-Jin Kim

Department of Radiology, Yonsei University College of Medicine, Seoul, Republic of Korea

* yelv@yuhs.ac

**Data Availability Statement:** All relevant data are within the paper and its Supporting Information files.

## Abstract

Herein, we assessed whether hepatobiliary phase (HBP) signal intensity (SI) can be used to differentiate HCC and non-HCC malignancies within LR-M observations. 106 LR-M patients based on LI-RADS v2018 who underwent gadoxetate-disodium magnetic resonance imaging and surgery from January 2009 to December 2018 were included. SI of LR-M observation on HBP was analyzed by two radiologists and categorized into dark, low and iso-to-high groups. Tumor was classified as dark when more than 50% of tumor showed hypointensity compared to spleen, as low when more than 50% of tumor showed hyperintensity compared to spleen but hypointensity compared to liver parenchyma, and as iso-to-high if there was even a focal iso-intensity or hyperintensity compared to liver parenchyma. Analysis of clinicopathological factors and association between imaging and histology was performed. Out of 106 LR-M, 42 (40%) were showed dark, 61 (58%) showed low, and 3 (3%) showed iso-to-high SI in HBP. Three iso-to-high SI LR-M were HCCs ($P$ = 0.060) and their major histologic differentiation was Edmondson grade 1 ($P$ = 0.001). 43 out of 61 (71%) low SI LR-M were iCCA or cHCC-CCA ($P$ = 0.002). Inter-reader agreement of HBP SI classification was excellent, with a kappa coefficient of 0.872. LR-M with iso-to-high SI in HBP is prone to being HCC while LR-M with low SI in HBP is prone to being tumor with fibrous stroma such as iCCA and cHCC-CCA. Classification of LR-M based on HBP SI may be a helpful method of differentiating HCC from non-HCC malignancies.

## Introduction

The CT/MRI liver imaging reporting and data system (LI-RADS) includes a special category, LI-RADS M (LR-M) for observations that are probably or definitely malignant but not necessarily hepatocellular carcinoma (HCC) [1]. The aim of this category, when first introduced, was to maintain the specificity of LR-5 (definitely HCC) without losing the sensitivity to detect malignancies including HCC with atypical imaging features, intrahepatic mass forming

**Funding:** Y.E.C. received faculty research grant of Yonsei University College of Medicine (No. 6-2019-0118). The funder did not play any role in the study design, data collection and analysis, decision to publish or preparation of the manuscript.

**Competing interests:** The authors have declared that no competing interests exist.

cholangiocarcinoma (iCCA) and combined hepatocellular cholangiocarcinoma (cHCC-CCA) [1, 2]. Although new explicit LR-M criteria have been introduced through LI-RADS v2017 (same in v2018) including targetoid apperance and several nontargetoid imaging features, the diagnostic performance of LR-M features for non-HCC malignancy has been variable with reported sensitivity of 9–83% and specificity of 69–97% [3–5]. Not only does the ambiguous criteria of LR-M makes it susceptible to the subjectivity of each radiologist but also the heterogeneous group of disease entities given this category makes it difficult for an accurate imaging prediction of the likely etiology of LR-M observation [6].

However, differential diagnosis of HCC from non-HCC malignancies on imaging is critical because pathologic confirmation is not always madated before instituting treatment in case of HCC and also because HCC differs from non-HCC malignancies such as cHCC-CCA and iCCA with regards to possible candidacy for liver transplantation and prediction of prognosis [7, 8]. In such case, it would be important to accurately categorize LR-M HCCs with atypical imaging features as definitely HCC in patients with Barcelona Clinic Liver Cancer (BCLC) stage 0/A and Child Pugh class A, who are eligible and can benefit curative treatment from liver transplantation [9–11]. Likewise, a more accurate image prediction of non-HCC malignancy within LR-M observations by differentiating non-HCC malignancies from HCC with atypical imaging features may help narrow patients in need and urgency of biopsy. Either way, a more accurate diagnosis of HCC or non-HCC malignancy among LR-M in patients of high risk of HCC holds mutual clinical significance for both groups. In addition, most of the previous studies [12–14] regarding LR-M observations have focused on imaging findings that can differentiate iCCA or cHCC-CCA from HCC among LR-M observations.

For diagnosing HCC, the use of gadoxetic acid-enhanced MRI is favored over extracellular agent (ECA)-enhanced MRI in East Asia since maximizing the sensitivity of HCC diagnosis is justified by the greater use of locoregional therapies such as ablation and transarterial chemoembolization, and the detection of HCC can be improved by the use of hepatobiliary phase of gadoxetic acid-enhanced MRI [15–17]. To our knowledge, while there have been studies analyzing tumor serum markers, imaging findings and deep learning [18, 19], no prior study has performed a quantitative assessment of the hepatobiliary phase signal intensity in order to differentiate a LR-M observation. Hence, the purpose of this study was to investigate whether hepatobiliary phase (HBP) signal intensity can be used to differentiate HCC and non-HCC malignancies within LR-M observations. In addition, image-histologic correlation was performed to provide histopathologic basis for the image manifestation and provide rationale to our criteria.

## Materials and methods

### Study population

This retrospective study was approved by our institutional review board of Yonsei University College of Medicine, Severance Hospital (IRB No. 2020-3696-001) and the requirement for patient consent was waived. Using electronic medical records, patients with underlying liver cirrhosis or chronic B-viral hepatitis who underwent gadoxetate-disodium enhanced MRI between January 2009 and December 2018 for the evaluation of a focal hepatic observation were identified. Patients who (1) underwent surgical resection within 6 months from date of MRI exam, (2) had not previously been treated for hepatic observation prior to MRI study, and (3) were pathologically diagnosed via surgery were included. Likewise, patients who (1) had poor MR image quality and (2) did not have all required images of MRI protocol were excluded from analysis. Based on these inclusion and exclusion criteria, 1,286 hepatic observations were eligible for study. The MRI data, surgical notes and pathology reports for the largest

observation in these patients were retrospectively reviewed. Two radiologists classified these observations according to LI-RADS v2018 in consensus (1), and LR-TIV and LR-1 to 5 observations were excluded, leaving 107 LR-M observations. According to LI-RADS v2018, LR-M is assigned to either targetoid mass or nontargetoid mass with one of infiltrative appearance, marked diffusion restriction, necrosis or severe ischemia and other feature that a radiologist judges to suggest a non-HCC malignancy (1).

Among 107 LR-M observations, one observation was excluded since hepatobiliary phase signal intensity of the observation could not be compared to the signal intensity of the spleen due to splenectomy status.

Clinical information and laboratory data of final 106 LR-M observations were then retrospectively reviewed and included the following: patient demographics, cause of chronic liver disease, serum levels of aspartate transaminase, alanine transaminase, total bilirubin, albumin, prothrombin time, platelets, α-fetoprotein, protein induced by vitamin K absence (PIVKA)-II, carbohydrate antigen 19–9 (CA 19–9), and carcinoembryonic antigen (CEA).

## MR imaging techniques

All patients underwent MRI exam via 3.0-T Magnetom Trio Tim (Siemens Medical Solutions, Erlangen, Germany), Intera Achieva or Ingenia (Philips Medical Systems, Best, the Netherlands), or Discovery MR750w MRI unit (GE Medical Systems, Waukesha, Wis). Dynamic liver MRI was performed using 10mL of gadoxetate-disodium (Primovist; Bayer AG, Berlin, Germany) at an injection rate of 1mL/sec, followed by 20mL of 0.9% saline chaser at the same flow rate (Spectris Solaris MR Injection System; Medrad, Warrendale, PA) [20]. T1 weighted 3D gradient-echo hepatobiliary phase (HBP) was obtained 20 minutes after contrast agent injection. Other MRI sequences included in routine dynamic liver MRI are written in the S1 Text.

## MR image analysis

One board certified radiologist and a senior radiology resident reviewed MR images using picture archiving and communication system (PACS) (Centricity Radiology RA 1000; GE Healthcare, Chicago, IL). Both reviewers were blinded to patient's clinical information and tumor histopathologic features. Tumor was classified in the dark group when more than 50% of tumor area showed hypointensity compared to spleen, in the low group when more than 50% of tumor area showed hyperintensity compared to spleen but hypointensity compared to liver parenchyma, and in the iso-to-high group if there was even a focal iso-intensity or hyperintensity compared to liver parenchyma on visual insepction in hepatobiliary phase image [21]. When equivocal on visual inspection, region of interest (ROI) was drawn on tumor, spleen and liver parenchyma to quantify and compare the signal intensities.

## Histopathology

Final diagnosis of hepatic observation and status of non-tumor liver parenchyma including presence of cirrhosis were extracted from pathology reports. For HCC, size, architectural pattern, variant/subtype and major histologic differentiation based on the nuclear grading scheme proposed by Edmondson and Steiner [22] were recorded. As for non-HCC malignancies, size and major histologic differentiation (well/moderate/poor/undifferentiated) were recorded. Presence of tumor necrosis (>5%), percentage of tumor necrosis in gross specimen, capsular formation status, and microvascular invasion status were recorded for all tumors.

## Statistical analysis

Inter-reader agreement was expressed by Cohen's kappa coefficient. A kappa statistic of 0.8–1.0 was considered excellent agreement, 0.6–0.89 good agreement, 0.40–0.59 moderate agreement, 0.2–0.39 fair agreement and 0–0.19 poor agreement. To compare features of HCC and non-HCC malignancies, we used Mann-Whitney U test for continuous variables and the $X^2$ or Fisher exact test for categorical variables. The association analyses of hepatobiliary phase signal intensity group versus tumor group and histopathologic findings were performed by calculating the Pearson's correlation coefficients and $p$ values. Bonferroni correction was used for post hoc multiple comparisons for all statistical analyses. Two-sided $p$ values <0.05 were considered as statistically significant. All statistical analyses were performed using R software (version 3.4.0; The R Foundation for Statistical Computing, Vienna, Austria).

## Results

### Diagnostic performance of combined LR-4 and LR-5, LR-5 and LR-M

Based on the eligibility criteria, final LR-categories were assigned to 1286 observations based on LI-RADS v2018 [1] and are summarized in Fig 1. Overall, the sensitivity and specificity of LR-4 and LR-5 combined, and LR-5 were 92.9% and 94.9%, and 71.6% and 98.3%, respectively (S1 Table).

### Baseline clinical characteristics of LR-M patients

Baseline patient characteristics are summarized in Tables 1 and 2. Our 106 LR-M patients comprised of 78 males and 28 females with mean age of 60 ± 11.5 years old. Chronic hepatitis B was the most predominant cause of underlying liver disease (80% of patients) and 48% had cirrhosis. Most patients (97%) were of Child Pugh class A and mean size of tumor was 38mm. The median duration between MRI and surgical pathology was 17 days.

Within 106 LR-M patients, 42 patients (40%) were HCCs and 64 patients (60%) were non-HCC malignancies. Most HCC patients (34/42, 81%) were of Barcelona Clinic Liver Cancer (BCLC) stage A and the rest were of BCLC stage 0.

Patients with non-HCC malignancies showed significantly older age (mean age: 62.6 vs. 56.2, $P = 0.005$), fewer underlying chronic hepatitis B background (73% vs. 91%, $P = 0.025$), lower alanine transaminase (ALT) (22.0 vs. 32.0, $P = 0.002$), α-fetoprotein (3.5 vs. 6.2, $P = 0.019$), PIVKA-II (26.0 vs. 47.0, $P = 0.003$) and higher CA19-9 (37.8 vs. 6.9, $P<0.001$) compared to patients with HCC (Table 1).

Subgroup analysis of non-HCC malignancies showed that 34% (22/64) had cHCC-CCA while 61% (39/64) had iCCA (Table 2). The remaining three patients had metastases: one ovarian cancer metastasis and two colon cancer metastases.

Subgroup analysis of LR-M observations showed that the significant differences in age, albumin, α-fetoprotein, and CA19-9 between HCC with atypical imaging features and non-HCC malignancies were mainly due to significant differences between HCC and iCCA: age ($P = 0.002$), underlying cirrhosis ($P = 0.014$), albumin ($P<0.001$), α-fetoprotein ($P<0.001$) and CA19-9 ($P<0.001$) (Table 2).

### Hepatobiliary phase signal intensity classification of HCC and non-HCC malignancies

Out of 106 LR-M observations, 42 observations (40%) were assigned dark, 61 observations (58%) were assigned low, and 3 observations (3%) were assigned iso-to-high signal intensities in hepatobiliary phase. Figs 2 and 3 show typical images of LR-M observations with dark, low

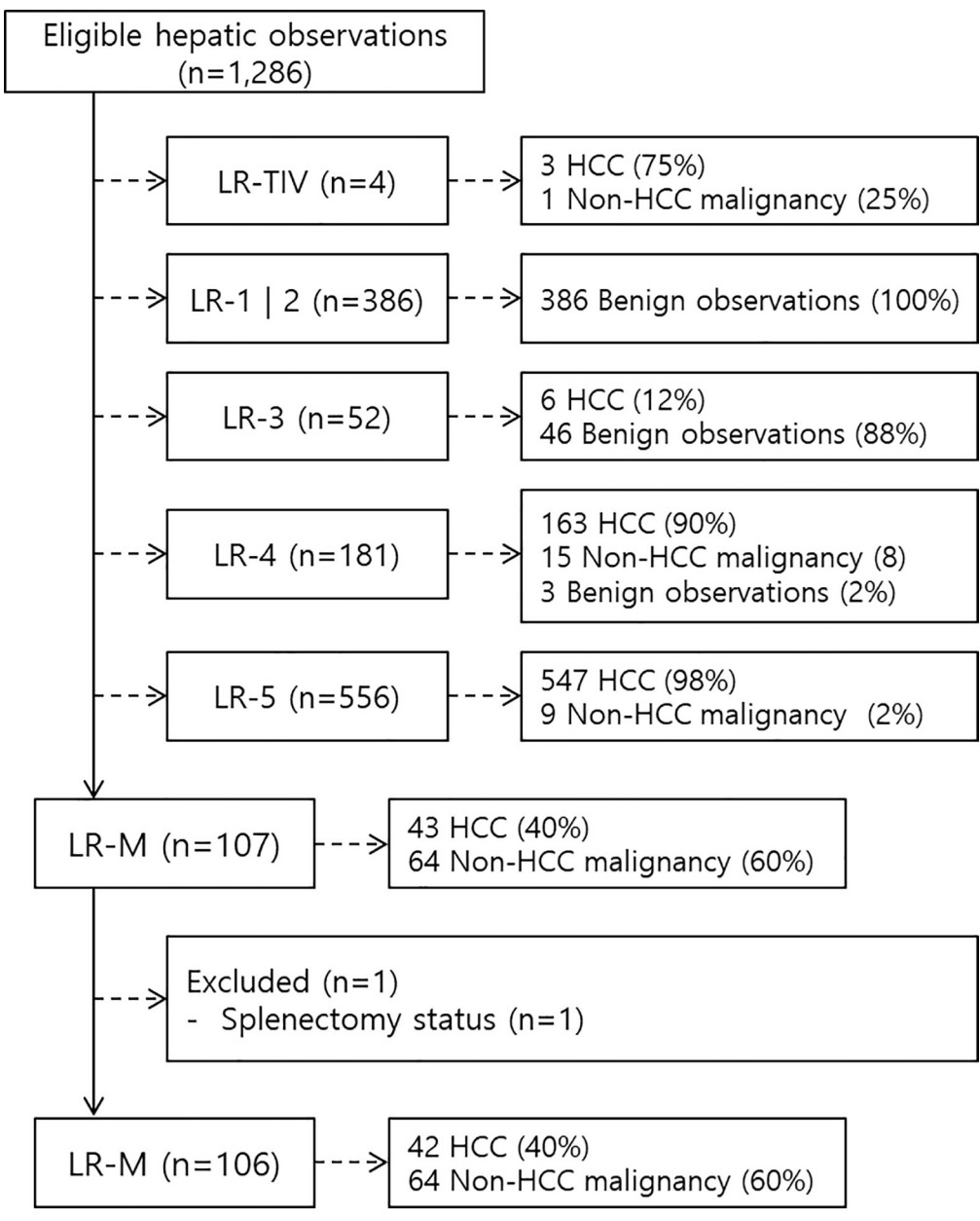

**Fig 1. Flow diagram of this study.**

and iso-to-high signal intensity in hepatobiliary phase. Nearly half of 42 dark observations (22, 52%) were found to be HCC while 24% (10/42) and 19% (8/42) were found to be iCCA and cHCC-CCA, respectively (Table 3). Significant associations between HCC and dark signal intensity over low signal intensity ($P = 0.036$) as well as between iCCA or cHCC-CCA or iCCA and low signal intensity over dark signal intensity ($Ps<0.05$) were found (Table 3). These associations were found significant under univariate analyses as well (S2 and S3 Tables). In case of iso-to-high observations, all three observations were found to be HCC although this association was not found to be statistically significant (P = 0.060) (S4 Table). Not a single cHCC-CCA, iCCA or metastasis was found to be iso-to-high in hepatobiliary phase.

**Table 1. Baseline characteristics of the included patients.**

| Variables | Total patients (n = 106, 100%) | HCC (n = 42, 40%) | Non-HCC malignancy (n = 64, 60%) | P-value |
|---|---|---|---|---|
| Age, years | 60.0 ± 11.5 | 56.2 ± 12.0 | 62.6 ± 10.6 | **0.005** |
| Sex | | | | 0.290 |
| Male | 78 (74) | 34 (81) | 44 (69) | |
| Female | 28 (26) | 8 (19) | 20 (31) | |
| Etiology | | | | |
| Hepatitis B | 85 (80) | 38 (91) | 47 (73) | **0.025** |
| Hepatitis C | 3 (3) | 0 (0) | 3 (5) | 0.270 |
| Alcohol | 12 (11) | 2 (5) | 10 (16) | 0.117 |
| NASH | 6 (6) | 2 (5) | 4 (6) | 0.999 |
| Child-Pugh Class | | | | 0.999 |
| A | 103 (97) | 41 (98) | 62 (97) | |
| B | 3 (3) | 1 (2) | 2 (3) | |
| Cirrhosis | 51 (48) | 25 (60) | 26 (41) | 0.088 |
| AST, IU/L | 27 (21–40) | 27 (21–44) | 27 (21–38) | 0.543 |
| ALT, IU/L | 25 (18–40) | 31 (24–44) | 22 (15–38) | **0.002** |
| Albumin, g/dL | 4.3 (4.1–4.6) | 4.5 (4.1–4.6) | 4.2 (3.9–4.5) | **0.034** |
| Total bilirubin (mg/dL) | 0.7 (0.5–1.0) | 0.8 (0.6–1.1) | 0.7 (0.5–0.9) | |
| PT, INR | 0.98 (0.94–1.03) | 0.98 (0.94–1.01) | 0.98 (0.93–1.05) | 0.837 |
| Platelets, 1000/μL | 208.4 ± 88.8 | 184.5 ± 61.0 | 224.8 ± 100.8 | **0.012** |
| AFP, IU/mL | 5.1 (2.7–51.9) | 6.2 (3.5–156.2) | 3.5 (2.5–14.9) | **0.019** |
| PIVKA-II, AU/mL | 31.0 (20.0–121.0) | 47.0 (24.0–282.0) | 26.0 (18.0–37.0) | **0.003** |
| CA 19–9, U/mL | 24.8 (8.4–123.0) | 6.9 (0.1–13.4) | 37.8 (15.3–392.0) | **<0.001** |
| CEA, ng/mL | 2.9 (1.7–4.8) | 2.9 (1.6–3.9) | 2.9 (1.8–6.0) | 0.272 |
| Observation size, mm | 38.6 ± 20.4 | 34.4 ± 16.1 | 41.4 ± 22.5 | 0.083 |
| Duration between MRI and surgical pathology, days | 17 (10–28) | 16 (8–24) | 18 (11–33) | 0.190 |
| BCLC stage 0 | | 8 (19) | | |
| BCLC stage A | | 34 (81) | | |

Numerical variables are expressed as median (interquartile range) or mean ± standard deviation, according to the result of normality test (Shapiro-Wilk test). Categorical variables are expressed as n (%). BCLC, Barcelona Clinic Liver Cancer; HCC, hepatocellular carcinoma; NASH, non-alcoholic steatohepatitis; AST, aspartate transaminase; ALT, alanine transaminase; AFP, α-fetorprotein; PIVKA, protein induced by vitamin K absence; CA 19–9, carbohydrate antigen 19–9; CEA, carcinoembryonic antigen; MRI, magnetic resonance imaging.

## Histopathologic correlation with hepatobiliary phase signal intensity classification

Histopathologic characteristics of LR-M observations based on hepatobiliary phase signal intensities are summarized in Table 4. In case of iCCA, cHCC-CCA and metastasis, no significant association was found between major histologic differentiation and dark, low and iso-to-high classification. However, in case of HCC, three observations which showed iso-to-high signal intensity were Edmondson grade 1 and this association was statistically significant (P = 0.001) (Fig 3). Moreover, HCCs with iso-to-high signal intensity showed pseudoglandular architectural pattern (P = 0.012). In addition, while not statistically significant, 7 out of 11 scirrhous HCC was found to show low signal intensity (P = 0.078) (Fig 4).

Presence of tumor necrosis (>5%), capsular formation, and microvascular invasion did not significantly differ among dark, low and iso-to-high groups. However, while the difference was nonsignificant, dark group showed larger necrotic percentage followed by low and iso-to-high group (P = 0.090). In case of scirrhous HCCs, those showing dark signal intensity had

**Table 2. Comparison of baseline characteristics of HCC, cHCC-CCA and CCA.**

| Variables | HCC (*n* = 42, 40%) | cHCC-CCA (*n* = 22, 21%) | iCCA (*n* = 39, 37%) | *P*-value | *P*-value[a] | *P*-value[b] | *P*-value[c] |
|---|---|---|---|---|---|---|---|
| Age, years | 56.2 ± 12.0 | 59.5 ± 9.6 | 64.8 ± 10.9 | **0.005** | 0.214 | 0.059 | **0.002** |
| Sex | | | | 0.520 | | | |
| Male | 34 (81) | 17 (77) | 26 (67) | | | | |
| Female | 8 (19) | 5 (23) | 13 (33) | | | | |
| Etiology | | | | | | | |
| Hepatitis B | 38 (91) | 17 (77) | 28 (72) | 0.075 | | | |
| Hepatitis C | 0 (0) | 0 (0) | 3 (8) | 0.071 | | | |
| Alcohol | 2 (5) | 3 (14) | 6 (15) | 0.237 | | | |
| NASH | 2 (5) | 2 (9) | 2 (5) | 0.757 | | | |
| Child-Pugh Class | | | | 0.874 | | | |
| A | 41 (98) | 21 (96) | 37 (97) | | | | |
| B | 1 (2) | 1 (5) | 1 (3) | | | | |
| Cirrhosis | 25 (60) | 13 (59) | 12 (31) | **0.019** | 0.941 | 0.057 | **0.014** |
| AST, IU/L | 27 (21–44) | 29 (23–58) | 27 (20–35) | 0.180 | | | |
| ALT, IU/L | 31 (24–44) | 27 (20–42) | 21 (13–34) | **0.002** | 0.325 | **0.036** | **<0.001** |
| Albumin, g/dL | 4.5 (4.1–4.6) | 4.2 (3.9–4.6) | 4.3 (4.0–4.4) | 0.133 | | | |
| Total bilirubin, mg/dL | 0.8 (0.6–1.1) | 0.8 (0.7–1.0) | 0.7 (0.4–0.9) | 0.070 | | | |
| PT, INR | 0.98 (0.94–1.01) | 1.01 (0.93–1.07) | 0.97 (0.92–1.03) | 0.683 | | | |
| Platelets, 1000/μL | 184.5 ± 61.0 | 155.0 ± 65.0 | 223 ± 107.6 | **<0.001** | 0.230 | **<0.001** | **<0.001** |
| AFP, IU/mL | 6.2 (3.5–156.2) | 11.6 (2.7–106.4) | 2.9 (2.2–4.5) | **0.001** | 0.967 | **0.002** | **<0.001** |
| PIVKA-II, AU/mL | 47.0 (24.0–282.0) | 28.5 (16.0–69.8) | 25.0 (18.0–34.0) | **0.008** | 0.048 | 0.402 | 0.004 |
| CA 19–9, U/mL | 6.9 (0.1–13.4) | 10.6 (8.7–36.7) | 95.0 (23.7–1478.0) | **<0.001** | **0.014** | **0.016** | **<0.001** |
| CEA, ng/mL | 2.9 (1.6–3.9) | 3.2 (1.8–4.9) | 2.7 (1.8–5.8) | 0.544 | | | |
| Observation size, mm | 34.4 ± 16.1 | 39.1 ± 16.5 | 36.8 ± 26.0 | 0.316 | | | |
| Duration between MRI and surgical pathology, days | 16 (8–24) | 19 (10–24) | 21 (12–37) | 0.119 | | | |

Numerical variables are expressed as median (interquartile range) or mean ± standard deviation, according to the result of normality test (Shapiro-Wilk test). Categorical variables are expressed as *n* (%). HCC, hepatocellular carcinoma; iCCA, intrahepatic mass-forming cholangiocarcinoma; cHCC-CCA, combined hepatocellular-cholangiocarcinoma; NASH, non-alcoholic steatohepatitis; AST, aspartate transaminase; ALT, alanine transaminase; AFP, α-fetorprotein; PIVKA, protein induced by vitamin K absence; CA 19–9, carbohydrate antigen 19–9; CEA, carcinoembryonic antigen; MRI, magnetic resonance imaging.
[a]Pair-wise comparison between HCC and cHCC-CCA.
[b]Pair-wise comparison between cHCC-CCA and iCCA.
[c]Pair-wise comparison between HCC and iCCA.

significantly higher mean tumor necrosis area compared to those showing low signal intensity (25.0 ± 21.2% vs. 2.5 ± 4.2%, *P* = 0.027).

## Inter-reader agreement of hepatobiliary phase signal intensity classification

Initially, reviewer 1 and reviewer 2 classified 44 (42%) and 43 (41%) as dark group, 59 (56%) and 60 (57%) as low group, and 3 (3%) and 3 (3%) as iso-to-high group, respectively (Table 5). The inter-reader agreement for hepatobiliary phase signal intensity classification was excellent, with a kappa coefficient of 0.872. Excellent inter-reader agreement was observed within HCC and within non-HCC malignancies with a kappa coefficient of 0.914 and 0.821, respectively.

## Discussion

In our study, LR-M observations that showed iso-to-high signal intensity in hepatobiliary phase were well-differentiated, Edmondson grade 1 HCCs with pseudoglandular architectural

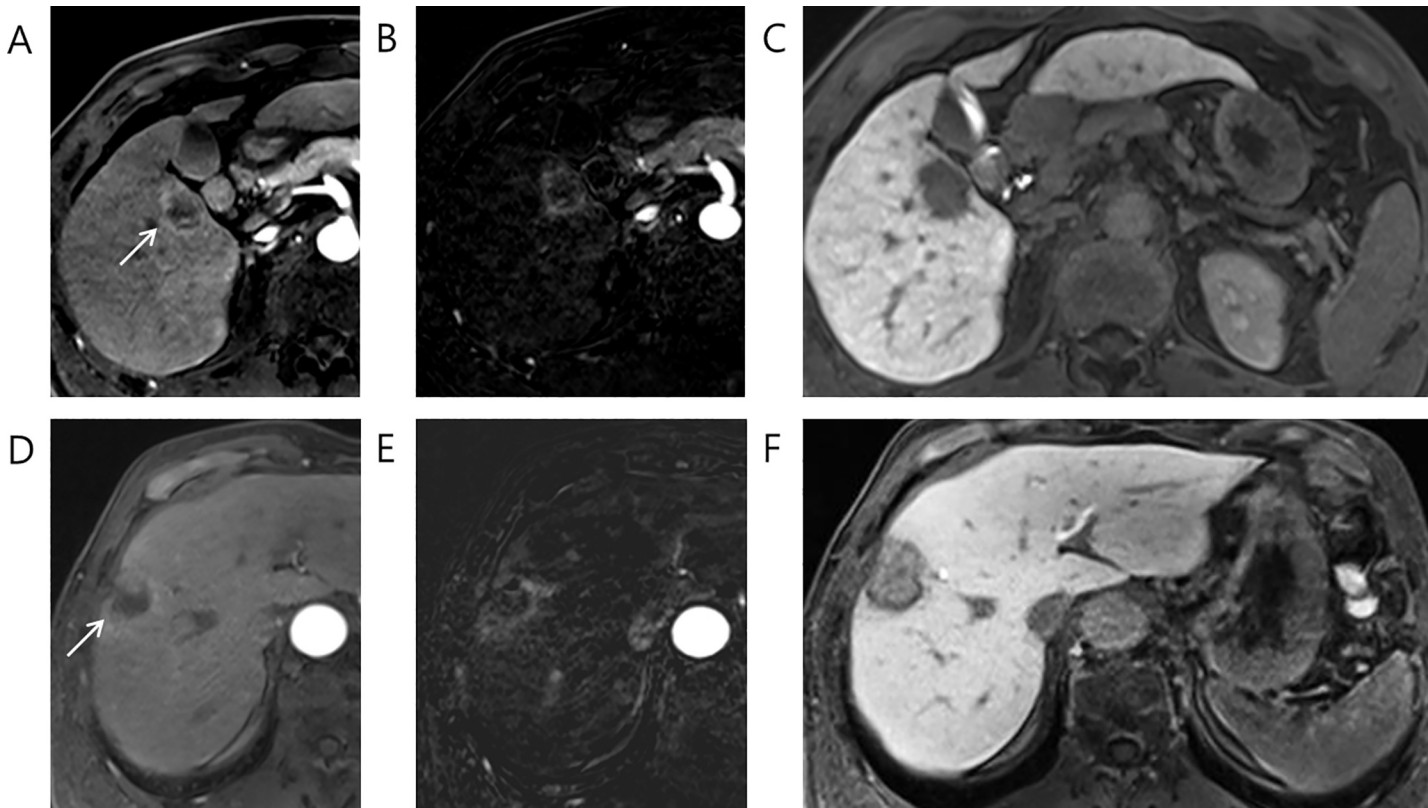

**Fig 2.** 46 year old male patient with Edmondson grade III HCC (A-C): rim APHE in (A) late arterial phase and (B) arterial subtraction image, (C) dark signal intensity in hepatobiliary phase. 74 year old male patient with intrahepatic mass-forming cholangiocarcinoma (D-F): rim APHE in (D) late arterial phase, and (E) arterial subtraction image, (F) low signal intensity in hepatobiliary phase.

pattern. On the other hand, nearly 70% (43 out of 61) of LR-M observations that showed low signal intensity in hepatobiliary phase were either cHCC-CCC or iCCA and this association was significant.

Imaging findings, especially that of hepatobiliary phase signal intensity of tumor may be useful in differentiating HCC with atypical imaging features from non-HCC malignancies

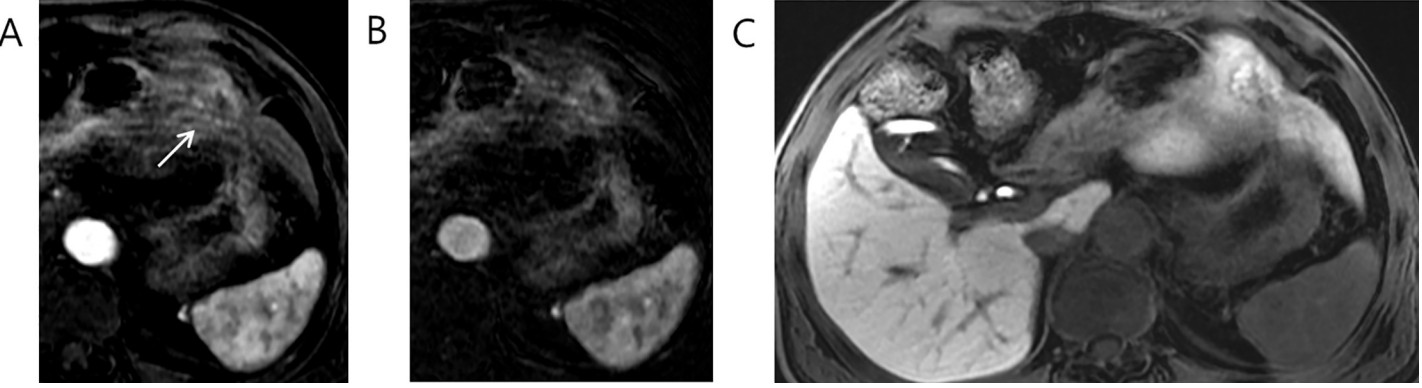

**Fig 3. 73 year old male patient with Edmondson grade I HCC.** Rim APHE is shown in (A) late arterial phase, (B) arterial subtraction image and (C) hepatobiliary phase. This hepatic observations shows iso-to-high signal intensity in hepatobiliary phase.

**Table 3. Hepatobiliary phase (HBP) signal intensities of HCC and non-HCC malignancies.**

| HBP Signal intensity | Dark | Low | Iso-to-High | P-value* | P-value[a] | P-value[b] | P-value[c] |
|---|---|---|---|---|---|---|---|
| **HCC (n = 42, 40%)** | 22 (52) | 17 (41) | 3 (7) | **0.004** | **0.036** | 0.081 | 0.242 |
| **cHCC-CCA (n = 22, 21%)** | 8 (36) | 14 (64) | 0 (0) | 0.595 | 0.635 | 0.999 | 0.999 |
| **iCCA (n = 39, 37%)** | 10 (26) | 29 (74) | 0 (0) | **0.026** | **0.045** | 0.245 | 0.999 |
| **cHCC-CCA or iCCA (n = 61, 58%)** | 18 (30) | 43 (69) | 0 (0) | **0.001** | **0.015** | 0.096 | 0.264 |
| **Metastasis (n = 3, 3%)** | 2 (67) | 1 (33) | 0 (0) | 0.615 | 0.565 | 0.999 | 0.999 |

Categorical variables are expressed as *n* (%). HBP, hepatobiliary phase; HCC, hepatocellular carcinoma; iCCA, intrahepatic mass-forming cholangiocarcinoma; cHCC-CCA, combined hepatocellular-cholangiocarcinoma.

Dark: signal intensity lower than spleen parenchyma.

Low: signal intensity higher than spleen parenchyma but lower than liver parenchyma.

Iso: signal intensity similar to liver parenchyma.

High: signal intensity higher than liver parenchyma.

*P-value calculated via $X^2$- test or Fisher's exact test comparing three groups (signal intensity) for each malignancy.

[a]Pairwise comparison between dark group and low group.

[b]Pairwise comparison between low group and iso-to-high group.

[c]Pairwise comparison between dark group and iso-to-high group.

among LR-M observations. Importantly, LR-M observations that showed iso-to-high signal intensity in hepatobiliary phase were HCCs. LR-M observations showing low signal intensity in hepatobiliary phase were more significantly associated with cHCC-CCA or iCCA. As for LR-M observations showing dark signal intensity in hepatobiliary phase, while significant association was found between dark LR-M and HCC, nearly 40% of dark LR-M also comprised of either cHCC-CCA or iCCA, making it a difficult differentiator of HCC from non-HCC malignancies.

In general, tumor signal intensity in hepatobiliary phase is known to decrease significantly during multistep hepatocarcinogenesis with worse histologic differentiation [23–25]. Consistent with previous studies, the number of iso-to-high signal intensity observations in hepatobiliary phase was highest in well-differentiated HCCs [23–26]. Expression of organic anion-transporting polypeptide 8 (OATP8), which is the most probable uptake transporter of gadoxetic acid, is reported to significantly decrease during multistep hepatocarcinogenesis due to increased expression of hepatocyte nuclear factor 3$\beta$ (HNF3$\beta$) [25, 27], which may explain why iso-to-high signal intensity HCCs were confirmed as well-differentiated HCC. Moreover, iso-to-high signal intensity HCCs were more significantly associated with pseudoglandular architectural pattern consistent with result of a previous study [24, 28]. Overexpression of OATP8 is thought to contribute to bile overproduction because OATP8 can also take up bile acid component, causing pseudoglandular proliferation with bile plugs and secondary dilatation of bile canaliculi [29].

Likewise, significant association between cHCC-CCA or iCCA and low signal intensity in hepatobiliary is consistent with results of previous studies [30–32]. Such low signal intensity in hepatobiliary phase is thought to relate to the presence of abundant stromal fibrosis in iCCA and cholangiocarcinoma component of cHCC-CCA, which causes extracellular accumulation of contrast agent through large interstitial spaces [31, 33].

Similarly, scirrhous HCC is known to exhibit fibrous tumor stroma generated by cancer-associated fibroblasts and peritumoral myofibroblasts through cross-talk with HCC cells [34]. In our study, more than half (58%) of scirrhous HCCs showed low signal intensity consistent with previous studies [35, 36]. Similar to iCCA and cHCC-CCA, scirrhous HCCs that did not show low signal intensity in hepatobiliary phase showed dark signal intensities. Previously,

**Table 4. Histopathologic characteristics of malignancies based on HBP signal intensities.**

| Tumor pathology | Data available | Dark | Low | Iso-to-High | P-value |
|---|---|---|---|---|---|
| **HCC (n, %)** | | 22 (52) | 17 (40) | 3 (7) | |
| Size (mm) | 42 | 35.3 ± 15.9 | 33.5 ± 18.6 | 32.2 ± 4.3 | 0.921 |
| Architectural pattern: | 42 | | | | |
| Trabecular | | 22 | 16 | 3 | 0.476 |
| Pseudoglandular | | 4 | 4 | 3 | **0.012** |
| Compact | | 1 | 4 | 0 | 0.220 |
| Histologic type | 42 | | | | |
| Classical | | 13 | 9 | 3 | 0.880 |
| Macrotrabecular-massive variant | | 2 | 0 | 0 | 0.566 |
| Scirrhous variant | | 4 | 7 | 0 | 0.078 |
| Lymphoepithelioma-like variant | | 3 | 0 | 0 | 0.397 |
| Sarcomatoid variant | | 0 | 1 | 0 | 0.476 |
| Major histologic differentiation | | | | | |
| Grade I / II/ III or IV * | 42 | 0 / 16 / 6 | 0 / 12 / 5 | 3 / 0 / 0 | **0.001** |
| **Intrahepatic mass-forming cholangiocarcinoma, iCCA, (n, %)** | | 10 (26) | 29 (74) | 0 (0) | |
| Size (mm) | 39 | 46.5 ± 27.8 | 42.2 ± 25.4 | | 0.655 |
| Major histologic differentiation | | | | | |
| Well / moderate / poor / undifferentiated | 38 | 2 / 6 / 1 / 0 | 5 / 21 / 3 / 0 | 0 / 0 / 0 / 0 | 0.429 |
| **Combined hepatocellular cholangiocarcinoma, cHCC-CCA (n, %)** | | 8 (36) | 14 (64) | 0 (0) | |
| Size (mm) | 22 | 42.2 ± 22.5 | 37.3 ± 12.5 | | 0.583 |
| Major histologic differentiation | | | | | |
| Well / moderate / poor / undifferentiated | 22 | 1 / 5 / 3 | 1 / 7 / 4 / 1 | 0 / 0 / 0 / 0 | 0.892 |
| **Metastasis (n, %)** | | 2 (67) | 1 (33) | 0 (0) | |
| Size (mm) | | 29.2 ± 16.7 | 40.0 ± 0 | | 0.691 |
| Major histologic differentiation | | | | | |
| Well / moderate / poor / undifferentiated | 3 | 0 / 1 / 1 / 0 | 0 / 1 / 0 / 0 | 0 / 0 / 0 / 0 | 0.667 |
| **Total patients (n, %)** | | | | | |
| Tumor necrosis (>5%) | | | | | |
| Absent / Present | 105 | 14 / 28 | 26 / 34 | 1 / 2 | 0.694 |
| Tumor necrosis area, % | 105 | 19.0 ± 23.9 | 10.9 ± 15.1 | 6.7 ± 7.6 | 0.090 |
| Capsular formation | | | | | |
| Absent / Partial / Complete | 102 | 22 / 16 / 2 | 51 / 4 / 4 | 0 / 2 / 1 | 0.451 |
| Microvascular invasion | | | | | |
| Absent / Present | 102 | 10 / 30 | 23 / 36 | 2 / 1 | 0.144 |

*Histologic differentiation of HCC is based on Edmondson grade.

studies on iCCA have reported heterogeneous tumor enhancement pattern in hepatobiliary phase to be attributed to the amount and density of fibrous component [37], timing of hepatobiliary phase and predominance of necrosis over fibrosis [38]. Consistent with these findings, the mean tumor necrosis area was higher in scirrhous HCCs showing dark signal intensities compared to those showing low signal intensities in our study. Importantly, however, combined together, iCCA, cHCC-CCA and scirrhous HCC comprised 82% of all low signal intensity LR-M observations.

There are some limitations in our study. First, this study may have a selection bias due to its retrospective design and inclusion of treatment-naïive patients with pathologically confirmed hepatic observations. However, we only accepted pathologically diagnoses as reference

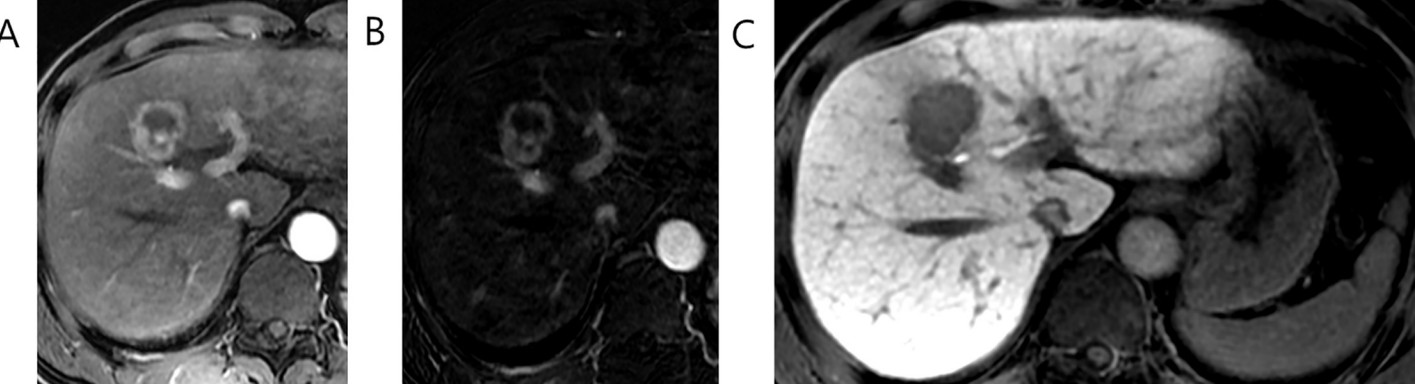

**Fig 4. 46 year old male patient with scirrhous HCC.** This observation shows rim APHE in (A) late arterial phase, (B) arterial subtraction image and (C) low signal intensity in hepatobiliary phase.

standards as imaging-histologic correlation is crucial for our analysis and because non-HCC malignancy such as cHCC-CCA has the potential to be misinterpreted as HCC based on imaging finding alone. Second, as we included pathologically confirmed cases, all of our patients were of Child Pugh class A or B and all HCC patients were of BCLC stage 0/A for whom diagnosis of HCC within LR-M observation would serve a clinical significance as liver transplantation is one method of curative treatment. However, the enhancement of liver parenchyma in hepatobiliary phase can be affected by hepatic function [39] and thus whether our method of classification in hepatobiliary phase is still valid in patients with worse Child-Pugh class and BCLC stage needs further study. In addition, due to the nature of retrospective study, quality of hepatobiliary phase image was not judged at the time of image acquisition but instead at the time of MRI reading. However, if the image quality was not diagnostic, exam was repeated in daily practice and hence, HBP image was quality-controlled. Third, there has been some minor modification to imaging protocols during the time period. However, these modifications were not significant enough to influence interpretation. Fourthly, we used 10mL of gadoxetate-disodium regardless of patient's weight as part of our hospital protocol. Lastly,

**Table 5. Inter-reader agreement of HBP signal intensities.**

|  | Reviewer 1 | Reviewer 2 | *K*, kappa | *P*-value |
|---|---|---|---|---|
| **Total patients (n = 106)** |  |  | 0.872 | <0.001 |
| Dark | 44 (42) | 43 (41) |  |  |
| Low | 59 (56) | 60 (57) |  |  |
| Iso-to-High | 3(3) | 3 (3) |  |  |
| **HCC (*n = 42*, 40%)** |  |  | 0.914 | <0.001 |
| Dark | 24 (57) | 22 (52) |  |  |
| Low | 15 (36) | 17 (40) |  |  |
| Iso-to-High | 3 (7) | 3 (7) |  |  |
| **Non-HCC malignancies (*n = 64*, 60%)** |  |  | 0.821 | <0.001 |
| Dark | 20 (31) | 21 (33) |  |  |
| Low | 44 (69) | 43 (67) |  |  |
| Iso-to-High | 0 (0) | 0 (0) |  |  |

A kappa statistic of 0.8–1.0 is considered excellent agreement, 0.6–0.79 good agreement, 0.40–0.59 moderate agreement, 0.2–0.39 fair agreement and 0–0.19 poor agreement.

image-histologic correlation of fibrous stroma of LR-M observations could not be performed as this information was not provided in routine pathology report.

In conclusion, LR-M observation showing iso-to-high signal intensity in hepatobiliary phase may be well-differentiated HCC while LR-M observations showing low signal intensity in hepatobiliary phase may be tumor with fibrous stroma such as iCCA, cHCC-CCA, or scirrhous HCC. Thus, classification of LR-M observations based on hepatobiliary phase signal intensity may be helpful in differentiating HCC with atypical imaging features from non-HCC malignancies.

## Supporting information

**S1 Text. MRI sequences included in liver dynamic MRI.**
(DOCX)

**S1 Table. Sensitivity, specificity, positive predictive value (PPV), negative predictive value and accuracy for HCC based on LI-RADS v2018 of eligible hepatic observations.**
(DOCX)

**S2 Table. $X^2$-test result HCC vs. dark signal intensity group in hepatobiliary phase.**
(DOCX)

**S3 Table. $X^2$-test result iCCA or cHCC-CCA vs. low signal intensity group in hepatobiliary phase.**
(DOCX)

**S4 Table. $X^2$-test result HCC vs. iso-to-high signal intensity group in hepatobiliary phase.**
(DOCX)

## Author Contributions

**Conceptualization:** Yong Eun Chung.

**Data curation:** Jae Hyon Park, Yong Eun Chung.

**Formal analysis:** Jae Hyon Park, Yong Eun Chung.

**Funding acquisition:** Yong Eun Chung.

**Investigation:** Yong Eun Chung.

**Methodology:** Yong Eun Chung.

**Project administration:** Yong Eun Chung.

**Resources:** Yong Eun Chung.

**Software:** Yong Eun Chung.

**Supervision:** Yong Eun Chung.

**Validation:** Jae Hyon Park, Yong Eun Chung, Jin-Young Choi.

**Visualization:** Yong Eun Chung.

**Writing – original draft:** Jae Hyon Park, Yong Eun Chung.

**Writing – review & editing:** Jae Hyon Park, Yong Eun Chung, Nieun Seo, Jin-Young Choi, Mi-Suk Park, Myeong-Jin Kim.

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
