## [Decision Letter · Decision Letter 0]

27 May 2021

PONE-D-21-02630

Hepatobiliary phase signal intensity: a potential method of diagnosing HCC with atypical imaging features among LR-M observations

PLOS ONE

Dear Dr. Chung,

Thank you for submitting your manuscript to PLOS ONE. After careful consideration, we feel that it has merit but does not fully meet PLOS ONE’s publication criteria as it currently stands. Therefore, we invite you to submit a revised version of the manuscript that addresses several points raised during the review process. the topic is original and the data of interest, some improvments in the form or the methods of analysis are requested.

We look forward to receiving your revised manuscript.

Kind regards,

Isabelle Chemin, PhD

Academic Editor

PLOS ONE

Journal Requirements:

Reviewers' comments:

Reviewer's Responses to Questions

**Comments to the Author**

1. Is the manuscript technically sound, and do the data support the conclusions?

Reviewer #1: Partly

2. Has the statistical analysis been performed appropriately and rigorously? 

Reviewer #1: Yes

3. Have the authors made all data underlying the findings in their manuscript fully available?

Reviewer #1: Yes

4. Is the manuscript presented in an intelligible fashion and written in standard English?

Reviewer #1: No

5. Review Comments to the Author

Reviewer #1: This paper aims to assess whether hepatobiliary phase signal intensity can be used to differentiate HCC and non-HCC malignancies within LR-M observations.

The study looks original even though a larger sample could have made significant data for HCC with iso-to-high signal.

- English needs to be revised: there are many typos (for example, lines 97, 193 and 267)

Introduction

-“the diagnostic performance of LR-M for non-HCC malignancy has been variable” This is too vague. Please provide %

- I suggest you to discuss here or in the discussion section about these recent papers to highlight that the same clinical issue has been recently investigated in different ways (“Recently, some studies …..[”Oestmann et al, Jiang H et al”]) including evaluation or serum markers and deep learning, but that to your knowledge no prior study performed a quantitative assessment on HBP.

Oestmann PM, Wang CJ, Savic LJ, Hamm CA, Stark S, Schobert I, Gebauer B, Schlachter T, Lin M, Weinreb JC, Batra R, Mulligan D, Zhang X, Duncan JS, Chapiro J. Deep learning-assisted differentiation of pathologically proven atypical and typical hepatocellular carcinoma (HCC) versus non-HCC on contrast-enhanced MRI of the liver. Eur Radiol. 2021 Jan 6. doi: 10.1007/s00330-020-07559-1.

Jiang H, Song B, Qin Y, Chen J, Xiao D, Ha HI, Liu X, Oloruntoba-Sanders O, Erkanli A, Muir AJ, Bashir MR. Diagnosis of LI-RADS M lesions on gadoxetate-enhanced MRI: identifying cholangiocarcinoma-containing tumor with serum markers and imaging features. Eur Radiol. 2021 Jun;31(6):3638-3648. doi: 10.1007/s00330-020-07488-z. Epub 2020 Nov 27. PMID: 33245494.

- You investigated this topic just on gadoxetate-disodium MRI. I would suggest to add a paragraph on why this contrast agent is often used in clinical practice in cirrhotic population. Please keep in mind that what happens in Eastern and Western countries may not be similar. As such, I would encourage to write a specific sentence highlighting the benefit of this contrast agent in this cohort of patients (see references

Zech CJ, Ba-Ssalamah A, Berg T, Chandarana H, Chau GY, Grazioli L, Kim MJ, Lee JM, Merkle EM, Murakami T, Ricke J, B Sirlin C, Song B, Taouli B, Yoshimitsu K, Koh DM. Consensus report from the 8th International Forum for Liver Magnetic Resonance Imaging. Eur Radiol. 2020 Jan;30(1):370-382. as specific guidelines

Vernuccio F, et al. AJR Am J Roentgenol. 2019 Aug;213(2):W57-W65. doi: 10.2214/AJR.18.20979. for its role in the diagnostic performance in indeterminate lesions

Materials and methods

Study cohort

- you included “patients with underlying liver cirrhosis or chronic B-viral hepatitis”. Please consider that LI-RADS includes also patients with prior HCC. Please be consistent with LI-RADS criteria

- “T1 weighted 3D gradient-echo hepatobiliary phase (HBP) was obtained 20 minutes after contrast agent injection." Has the hepatobiliary phase been judged appropriate by a radiologist for all examinations? If so, specify it. Indeed, Image F of Figure 2 is not an adequate hepatobiliary phase, which compromises signal intensity assessment. Please see these criteria in this paper and mention it: Khouri Chalouhi C, et al. Eur Radiol. 2019 Jun;29(6):3090-3099.

- Did you use 10 ml of gadoxetate-disodium regardless of the patient's weight?

Results

- You should not emphasize or specifically focus on the 3 HCC with iso-to-high signal as this does not represent a relevant data (too few patients), rather more centrality should be given to the other significative data obtained.

-“ Nearly half of 42 dark observations (22, 51%) were found to be while”. The word HCC is missing.

- “Out of 106 LR-M observations, 42 observations (42%) were assigned dark, 61 observation (58%) were assigned low, and 3 observations (3%)” Percentages are incorrect: 42/106 (39.6%), 61/106 (57.54%), 2/106 (2.8%)

“In case of iso-to-high observations, all three observations were found to be HCC although this association was not found to be statistically significant (P=0.060) (S4 Table). “ Was the HBP phase adequate in these patients? I am pretty sure it was not adequate in one of this case as it is shown in Figure 2 and this is not an adequate HBP

Tables

Please check ALL the % in the tables

Discussion

- All the first sentence of the discussion needs to be deleted as in one of the case (Fig 2) HBP was inadequate and because these are only 3 cases and you can not draw conclusions on this. I would highlight other interesting points.

-multisetep—>multistep

-“Likewise, significant association between cHCC-CCA 275 or iCCA and low signal intensity in

hepatobiliary is consistent with previous studies (25-27). Such low signal intensity in

hepatobiliary phase is thought to relate to the presence of abundant stromal fibrosis in iCCA

and cholangiocarcinoma component of cHCC-CCA, which causes extracellular accumulation

of contrast agent through large interstitial spaces (26, 28)."

- It is not exactly this way. When you have this abundant stroma you may have a cloud appearance as shown in this paper: Insights Imaging. 2021 Jan 12;12(1):8. doi: 10.1186/s13244-020-00928-w.

Conclusion must be rewritten without focusing on those 3 iso-to-high HBP lesions for the reasons mentioned above.

6. PLOS authors have the option to publish the peer review history of their article (what does this mean?). If published, this will include your full peer review and any attached files.

Reviewer #1: No

---

## [Author Response · Author response to Decision Letter 0]

27 Jun 2021

Review Comments to the Author

Reviewer #1: This paper aims to assess whether hepatobiliary phase signal intensity can be used to differentiate HCC and non-HCC malignancies within LR-M observations.

 The study looks original even though a larger sample could have made significant data for HCC with iso-to-high signal.

Comment #1 - English needs to be revised: there are many typos (for example, lines 97, 193 and 267)

We are sorry for the typos. We asked a native English speaker to review our revised manuscript.

Introduction

Comment #2. -“the diagnostic performance of LR-M for non-HCC malignancy has been variable” This is too vague. Please provide %

As suggested by Reviewer #1, we included the range of reported sensitivity and specificity of LR-M features and revised our sentence to read as follows: “..the diagnostic performance of LR-M features for non-HCC malignancy has been variable with reported sensitivity of 9-83% and specificity of 69-97%” (reference: Kim YY, Kim MJ, Kim EH, Roh YH, An C. Hepatocellular Carcinoma versus Other Hepatic Malignancy in Cirrhosis: Performance of LI-RADS Version 2018. Radiology. 2019;291(1):72-80). 

Comment #3 - I suggest you to discuss here or in the discussion section about these recent papers to highlight that the same clinical issue has been recently investigated in different ways (“Recently, some studies …..[”Oestmann et al, Jiang H et al”]) including evaluation or serum markers and deep learning, but that to your knowledge no prior study performed a quantitative assessment on HBP.

 Oestmann PM, Wang CJ, Savic LJ, Hamm CA, Stark S, Schobert I, Gebauer B, Schlachter T, Lin M, Weinreb JC, Batra R, Mulligan D, Zhang X, Duncan JS, Chapiro J. Deep learning-assisted differentiation of pathologically proven atypical and typical hepatocellular carcinoma (HCC) versus non-HCC on contrast-enhanced MRI of the liver. Eur Radiol. 2021 Jan 6. doi: 10.1007/s00330-020-07559-1.

 Jiang H, Song B, Qin Y, Chen J, Xiao D, Ha HI, Liu X, Oloruntoba-Sanders O, Erkanli A, Muir AJ, Bashir MR. Diagnosis of LI-RADS M lesions on gadoxetate-enhanced MRI: identifying cholangiocarcinoma-containing tumor with serum markers and imaging features. Eur Radiol. 2021 Jun;31(6):3638-3648. doi: 10.1007/s00330-020-07488-z. Epub 2020 Nov 27. PMID: 33245494.

As suggested by Reviewer #1, we included the above references and discussed them in the introduction as follows: “To our knowledge, while there have been studies analyzing tumor serum markers, imaging findings and deep learning (15, 16), no prior study has performed a quantitative assessment of hepatobiliary phase signal intensity in order to differentiate a LR-M observation.”

Comment #4 - You investigated this topic just on gadoxetate-disodium MRI. I would suggest to add a paragraph on why this contrast agent is often used in clinical practice in cirrhotic population. Please keep in mind that what happens in Eastern and Western countries may not be similar. As such, I would encourage to write a specific sentence highlighting the benefit of this contrast agent in this cohort of patients (see references

 Zech CJ, Ba-Ssalamah A, Berg T, Chandarana H, Chau GY, Grazioli L, Kim MJ, Lee JM, Merkle EM, Murakami T, Ricke J, B Sirlin C, Song B, Taouli B, Yoshimitsu K, Koh DM. Consensus report from the 8th International Forum for Liver Magnetic Resonance Imaging. Eur Radiol. 2020 Jan;30(1):370-382. as specific guidelines

 Vernuccio F, et al. AJR Am J Roentgenol. 2019 Aug;213(2):W57-W65. doi: 10.2214/AJR.18.20979. for its role in the diagnostic performance in indeterminate lesions

As suggested by Reviewer #1, we added sentences to provide basis for why we analyzed gadoxetic acid- enhanced MRI and also included the above references. 

 Materials and methods

 Study cohort

Comment #5 - you included “patients with underlying liver cirrhosis or chronic B-viral hepatitis”. Please consider that LI-RADS includes also patients with prior HCC. Please be consistent with LI-RADS criteria

We agree with Reviewer #1 that patients with prior HCC can be also evaluated using LI-RADS criteria. However, we chose to evaluate treatment-naive patients with chronic hepatitis B virus infection and/or liver cirrhosis similar to other previous published studies; yet, we made sure to acknowledge this under limitations. 

- Rhee H, Cho ES, Nahm JH, Jang M, Chung YE, Baek SE, et al. Gadoxetic acid-enhanced MRI of macrotrabecular-massive hepatocellular carcinoma and its prognostic implications. Journal of hepatology. 2021;74(1):109-21.

- An C, Kim DW, Park YN, Chung YE, Rhee H, Kim MJ. Single Hepatocellular Carcinoma: Preoperative MR Imaging to Predict Early Recurrence after Curative Resection. Radiology. 2015;276(2):433-43.

- Kim YY, Kim MJ, Kim EH, Roh YH, An C. Hepatocellular Carcinoma versus Other Hepatic Malignancy in Cirrhosis: Performance of LI-RADS Version 2018. Radiology. 2019;291(1):72-80.

Comment #6- “T1 weighted 3D gradient-echo hepatobiliary phase (HBP) was obtained 20 minutes after contrast agent injection." Has the hepatobiliary phase been judged appropriate by a radiologist for all examinations? If so, specify it. Indeed, Image F of Figure 2 is not an adequate hepatobiliary phase, which compromises signal intensity assessment. Please see these criteria in this paper and mention it: Khouri Chalouhi C, et al. Eur Radiol. 2019 Jun;29(6):3090-3099.

 - Did you use 10 ml of gadoxetate-disodium regardless of the patient's weight?

Yes, while the hepatobiliary phase image was not judged at the time of the image acquisition, image qualities were judged at the time of MRI reading and if not diagnostic, we repeated the exam (image acquisition) in daily practice. Since we performed a retrospective study, all the images were judged at the timing of MRI reading and if inadequate, exam was repeated; thus we believe an adequate quality control was done. We agree that Figure 2 does not appear as a typical HBP image. Sometimes, even when the patient does not have a biliary obstruction, HBP image can appear as Figure 2 even after repeat examination. Since we use both extracellular agent (ECA) and EOB contrast at our hospital, we also confirmed the type of contrast used and the timing of image aquisition since contrast injection. We can assure that the Figure 2 patient underwent EOB contrast MRI and the contrast is pooled in bile ducts in other axial images along z-axis.

As proof, below are the other HBP images of Figure 2 patient, and you can see that while the signal intensity of intrahepatic vessels are not lower than that of liver parenchyme, these images are indeed EOB contrast images since the contrast is pooled in gall bladder and bile ducts. These images were taken abour 20 minutes past time of contrast injection.

This same patient also underwent EOB contrast MRI after surgery (left lobectomy) and the HBP images were as follows: 

Similar to pre-operative EOB contrast MRI, the intrahepatic vessels did not show lower signal intensity than liver parenchyme in HBP, but the HBP was obtained when contrast is pooled in the biliary ducts.

Yes, we used 10mL of gadeoxetic-disodium regardless of patient’s weight as it is our hospital protocol

Results

Comment #7 - You should not emphasize or specifically focus on the 3 HCC with iso-to-high signal as this does not represent a relevant data (too few patients), rather more centrality should be given to the other significative data obtained.

We thank Reviewer #1 for his or her comment. We agree with Reviewer #1 that the number of HCC with iso to high HBP signal intensity is small, and while statistically significant, may not be representative. However, the number of HCCs with iso to high HBP signal intensity and without nonrim arterial hyperenhancement is reported to be extremely rare (about 0.7%, 2/304) among HCCs (Choi JW, Lee JM, Kim SJ, Yoon JH, Baek JH, Han JK, et al. Hepatocellular carcinoma: imaging patterns on gadoxetic acid-enhanced MR Images and their value as an imaging biomarker. Radiology. 2013;267(3):776-86). This is partly because most HCC with iso to high HBP signal intensity are categorized as LR-3, LR-4 and LR-5 and are rarely categorized as LR-M. However, while there are only three patients in our study, considering that these patients are BCLC stage 0/A and Child Pugh class A, they can be possible candiates of liver transplantation. In such sense, we do not believe that three out of 106 LR-M (~3%) is a small number. However, we understand Reviewer #1’s concern and thus we made sure to “tone down” the significance of these findings and revised our sentences in both results section and discussion section. 

Comment #8 -“ Nearly half of 42 dark observations (22, 51%) were found to be while”. The word HCC is missing.

We are deeply sorry for this mistake, and we have revised the above sentence. 

Comment #9 - “Out of 106 LR-M observations, 42 observations (42%) were assigned dark, 61 observation (58%) were assigned low, and 3 observations (3%)” Percentages are incorrect: 42/106 (39.6%), 61/106 (57.54%), 2/106 (2.8%)

We are deeply sorry for this mistake and revised the above % from 42% to 40%.

Comment #10 “In case of iso-to-high observations, all three observations were found to be HCC although this association was not found to be statistically significant (P=0.060) (S4 Table). “ Was the HBP phase adequate in these patients? I am pretty sure it was not adequate in one of this case as it is shown in Figure 2 and this is not an adequate HBP

We thank Reviewer #1 for his or her comment. As mentioned above, the HBP images were quality controlled in this retrospective study. In addition, Figure 3 instead of Figure 2 is an example of HBP iso to high LR-M observation and HBP was adequate in these patients (Figure 2 is an example of HBP dark and low group).

 Tables

Comment #11 Please check ALL the % in the tables

We have checked all the % in the tables and manuscript and revised accordingly. 

 Discussion

Comment #12 - All the first sentence of the discussion needs to be deleted as in one of the case (Fig 2) HBP was inadequate and because these are only 3 cases and you cannot draw conclusions on this. I would highlight other interesting points.

We thank Reviewer #1 for his or her comment. As already mentioned above, HBP was adequately assessed in our study, and Figure 2 is not an example of HBP iso to high but rather HBP dark to low group, and while three cases were obtained for iso to high group, their associations with well differentiated HCC and pseudoglandular pattern were statistically significant (Table 4). Similar to comment mentioned in Results section, we, however, understand Reviewer’s concern and thus revised our sentences to “tone-down” the significance of results obtained for HBP iso to high LR-M observations. 

Comment #13 -multisetep—>multistep

We have made the above correction as suggested by Reviewer #1. 

Comment #14 -“Likewise, significant association between cHCC-CCA 275 or iCCA and low signal intensity in

 hepatobiliary is consistent with previous studies (25-27). Such low signal intensity in

 hepatobiliary phase is thought to relate to the presence of abundant stromal fibrosis in iCCA

 and cholangiocarcinoma component of cHCC-CCA, which causes extracellular accumulation

 of contrast agent through large interstitial spaces (26, 28)."

 - It is not exactly this way. When you have this abundant stroma you may have a cloud appearance as shown in this paper: Insights Imaging. 2021 Jan 12;12(1):8. doi: 10.1186/s13244-020-00928-w.

We thank Reviewer #1 for his or her comment. The cloud appearance (also referred to as “EOB-cloud” in some studies) in which the central portion of the tumor shows delayed enhancement is the reason why we believe these tumors were grouped as “low” in HBP (lower than liver parenchyme enhancement but not as dark as those categorized as HBP “dark” group). Cloud appearance, as suggested by Reviewer #1 is due to stromal fibrosis, and in this paper (Jeong HT, Kim M-J, Chung YE, Choi JY, Park YN, Kim KW. Gadoxetate Disodium–Enhanced MRI of Mass-Forming Intrahepatic Cholangiocarcinomas: Imaging-Histologic Correlation. American Journal of Roentgenology. 2013;201(4):W603-W11), which performed an image-histology correlation, the authors state that “large interstitial spaces seen in stromal fibrosis may retain contrast agents… the target appearance is obtained… also with gadoxetate disodium (page W610)”. Based on this reference, we meant to state that tumors in HBP low group are categorized in this group due to the cloud appearance which may have a pathologic basis to the abovementioned retained contrast in interstitial spaces due to stromal fibrosis.

Comment #15 - Conclusion must be rewritten without focusing on those 3 iso-to-high HBP lesions for the reasons mentioned above.

- As suggested by Reviewer #1, we excluded our results on HBP iso-to-high LR-M observations in conclusion and mentioned only about HBP low LR-M observations.

---

## [Decision Letter · Decision Letter 1]

31 Jul 2021

PONE-D-21-02630R1

Hepatobiliary phase signal intensity: a potential method of diagnosing HCC with atypical imaging features among LR-M observations

PLOS ONE

Dear Dr. Chung,

Thank you for submitting your manuscript to PLOS ONE. After careful consideration, we feel that it has merit but does not fully meet PLOS ONE’s publication criteria as it currently stands. Therefore, we invite you to submit a revised version of the manuscript that addresses the remaining points raised during the review process.

We look forward to receiving your revised manuscript.

Kind regards,

Isabelle Chemin, PhD

Academic Editor

PLOS ONE

Journal Requirements:

Reviewers' comments:

Reviewer's Responses to Questions

**Comments to the Author**

1. If the authors have adequately addressed your comments raised in a previous round of review and you feel that this manuscript is now acceptable for publication, you may indicate that here to bypass the “Comments to the Author” section, enter your conflict of interest statement in the “Confidential to Editor” section, and submit your "Accept" recommendation.

Reviewer #1: (No Response)

2. Is the manuscript technically sound, and do the data support the conclusions?

Reviewer #1: Yes

3. Has the statistical analysis been performed appropriately and rigorously? 

Reviewer #1: Yes

4. Have the authors made all data underlying the findings in their manuscript fully available?

Reviewer #1: Yes

5. Is the manuscript presented in an intelligible fashion and written in standard English?

Reviewer #1: Yes

6. Review Comments to the Author

Reviewer #1: 1. In the abstract and in the main document please change “parenchyme“ to parenchyma

2. “Yes, we used 10mL of gadeoxetic-disodium regardless of patient’s weight as it is our hospital protocol ”

This needs to be added as a limitation because this is not what it is recommended in the drug leaflet

3. Regarding figure 2 I understand that the contrast agent was in the bile ducts but the lower signal of vessels compared to liver parenchyma is also a criterion to judge adequacy of the HBP. I know that this feature may lack even after 40 mins after contrast injection typically in patients with high bilirubin levels, but Figure 2 and the lack of all criteria for HBP adequacy may raise concern from the reader’s perspective. I suggest to identify a different case for Figure 2, if this is not too much burden for you. In case this is highly difficult, please feel free to disregard this comment

4. In figure 2 and 3 it is not needed to have an arrow for each figure part. Just leave it in the first image showing the lesion (2a, 2d and 3a) and delete it from the remaining.

7. PLOS authors have the option to publish the peer review history of their article (what does this mean?). If published, this will include your full peer review and any attached files.

Reviewer #1: No

---

## [Author Response · Author response to Decision Letter 1]

5 Aug 2021

Review Comments to the Author:

- We foremost thank the Reviewer for his or her thoughtful comments and for reviewing our manuscript. 

Reviewer #1: 1. In the abstract and in the main document please change “parenchyme“ to parenchyma

- As recommended, we changed the word “parenchyme” to “parenchyma” in the abstract and main manuscript. 

2. “Yes, we used 10mL of gadeoxetic-disodium regardless of patient’s weight as it is our hospital protocol ”

This needs to be added as a limitation because this is not what it is recommended in the drug leaflet

- As recommended, we included this as a limitation in the discussion section.

3. Regarding figure 2 I understand that the contrast agent was in the bile ducts but the lower signal of vessels compared to liver parenchyma is also a criterion to judge adequacy of the HBP. I know that this feature may lack even after 40 mins after contrast injection typically in patients with high bilirubin levels, but Figure 2 and the lack of all criteria for HBP adequacy may raise concern from the reader’s perspective. I suggest to identify a different case for Figure 2, if this is not too much burden for you. In case this is highly difficult, please feel free to disregard this comment

- We agree with the Reviewer that changing figure 2 example of low group would be better for the readers. As recommended, we modified Figure 2 using different case. 

4. In figure 2 and 3 it is not needed to have an arrow for each figure part. Just leave it in the first image showing the lesion (2a, 2d and 3a) and delete it from the remaining.

- As recommended, we removed arrows in Figure 2 and 3 and left it in first images (2a, 2d and 3a).

---

## [Editor Report · Decision Letter 2]

31 Aug 2021

Hepatobiliary phase signal intensity: a potential method of diagnosing HCC with atypical imaging features among LR-M observations

PONE-D-21-02630R2

Dear Dr. Chung,

We’re pleased to inform you that your manuscript has been judged scientifically suitable for publication and will be formally accepted for publication once it meets all outstanding technical requirements.

Kind regards,

Isabelle Chemin, PhD

Academic Editor

PLOS ONE
---

## [Editor Report · Acceptance letter]

3 Sep 2021

PONE-D-21-02630R2 

Hepatobiliary phase signal intensity: a potential method of diagnosing HCC with atypical imaging features among LR-M observations 

Dear Dr. Chung:

I'm pleased to inform you that your manuscript has been deemed suitable for publication in PLOS ONE. Congratulations! Your manuscript is now with our production department. 

Kind regards, 

on behalf of

Mrs Isabelle Chemin 

Academic Editor

PLOS ONE